# Modelling the Spatial and Temporal Dynamics of Paralytic Shellfish Toxins (PST) at Different Scales: Implications for Research and Management

**DOI:** 10.3390/toxins14110786

**Published:** 2022-11-11

**Authors:** Patricio A. Díaz, Carlos Molinet, Miriam Seguel, Edwin J. Niklitschek, Manuel Díaz, Gonzalo Álvarez, Iván Pérez-Santos, Daniel Varela, Leonardo Guzmán, Camilo Rodríguez-Villegas, Rosa I. Figueroa

**Affiliations:** 1Centro i~mar, Universidad de Los Lagos, Casilla 557, Puerto Montt 5290000, Chile; 2CeBiB, Universidad de Los Lagos, Casilla 557, Puerto Montt 5290000, Chile; 3Programa de Investigación Pesquera, Instituto de Acuicultura, Universidad Austral de Chile, Puerto Montt 5489001, Chile; 4Programa Integrativo, Centro Interdisciplinario para la Investigación Acuícola (INCAR), Concepción 4030000, Chile; 5Centro Regional de Análisis de Recursos y Medio Ambiente (CERAM), Universidad Austral de Chile, Puerto Montt 5110566, Chile; 6Facultad de Ciencias del Mar, Departamento de Acuicultura, Universidad Católica del Norte, Larrondo 1281, Coquimbo 1780000, Chile; 7Centro de Investigación y Desarrollo Tecnológico en Algas (CIDTA), Facultad de Ciencias del Mar, Universidad Católica del Norte, Larrondo 1281, Coquimbo 1780000, Chile; 8Centro de Investigación Oceanográfica COPAS Sur-Austral and COPAS COASTAL, Universidad de Concepción, Concepción 4030576, Chile; 9Centro de Investigaciones en Ecosistemas de la Patagonia (CIEP), Coyhaique 5951369, Chile; 10Centro de Estudios de Algas Nocivas (CREAN), Instituto de Fomento Pesquero (IFOP), Puerto Montt 5480000, Chile; 11Centro Oceanográfico de Vigo, Instituto Español de Oceanografía (IEO-CSIC), Subida a Radio Faro 50, 36390 Vigo, Spain

**Keywords:** *Alexandrium catenella*, paralytic shellfish toxins (PST), paralytic shellfish poisoning (PSP), detoxification dynamics, spatial scales, Chilean fjords

## Abstract

Harmful algal blooms, in particular recurrent blooms of the dinoflagellate *Alexandrium catenella*, associated with paralytic shellfish poisoning (PSP), frequently limit commercial shellfish harvests, resulting in serious socio-economic consequences. Although the PSP-inducing species that threaten the most vulnerable commercial species of shellfish are very patchy and spatially heterogeneous in their distribution, the spatial and temporal scales of their effects have largely been ignored in monitoring programs and by researchers. In this study, we examined the spatial and temporal dynamics of PSP toxicity in the clam (*Ameghinomya antiqua*) in two fishing grounds in southern Chile (Ovalada Island and Low Bay). During the summer of 2009, both were affected by an intense toxic bloom of *A. catenella* (up to 1.1 × 10^6^ cells L^−1^). Generalized linear models were used to assess the potential influence of different environmental variables on the field detoxification rates of PSP toxins over a period of 12 months. This was achieved using a four parameter exponential decay model to fit and compare field detoxification rates per sampling site. The results show differences in the spatial variability and temporal dynamics of PSP toxicity, given that greater toxicities (+10-fold) and faster detoxification (20% faster) are observed at the Ovalada Island site, the less oceanic zone, and where higher amounts of clam are annually produced. Our observations support the relevance of considering different spatial and temporal scales to obtain more accurate assessments of PSP accumulation and detoxification dynamics and to improve the efficacy of fisheries management after toxic events.

## 1. Introduction

The direct and negative impacts of harmful algal blooms (HABs) on human health and economic sectors, mainly those related to aquaculture and the shellfish industry, are well-established (e.g., review by [1,2,3]). Chile has been severely affected by HABs, especially those caused by the dinoflagellate species *Alexandrium catenella* (Whedon and Kofoid) Balech, responsible for paralytic shellfish poisoning (PSP) [4]. Blooms of *A. catenella* are more frequent and intense in the southernmost regions of the country (Aysén and Magallanes, see Appendix A; Figure A1) [5,6]. During a bloom, the paralytic shellfish toxins (PSTs) produced in dinoflagellate cells are consumed by plankton feeders (such as commercial bivalves), where they accumulate [7,8,9]. If the regulatory limit for PST accumulation in shellfish (80 µg STX eq. 100 g^−1^) is exceeded, the extraction and commercialization of bivalve mollusks is banned in order to protect consumer health [10,11].

The *A. catenella* outbreaks in southern Chile during the summers of 2009 and 2016 reached remarkably high cell densities [9,12,13], causing alarm regarding a possible recrudescence of PSP events of high intensity and spatial extent, as such blooms are also spread geographically [5,6,14,15,16]. Consequently, the need for appropriate measures to minimize losses and effectively predict recovery times gained new urgency in order to allow either partial (i.e., within “extractive windows”) or full exploitation of affected shellfish stocks. At smaller geographical scales, variability in local or regional oceanography or hydrography is critical to bloom location and timing [17,18]. Accordingly, the development of effective management actions must be based on knowledge of the spatial and temporal dynamics of PSP events at these scales. However, such information is, for the most part, lacking and the smallest scales, such as fishing grounds, have scarcely been addressed, even though they are the most consequential for shellfish fleets and the local shellfish industry. The elucidation of regional predictors relevant at a more practical scale would be of enormous value to the respective stakeholders.

Large-scale monitoring studies show that habitat heterogeneity and seasonal patterns are among the significant factors in predictions of the spatial and temporal dynamics of PST events in southern Chile. Habitat heterogeneity influences PST accumulation and field detoxification rates in natural shellfish populations [5,19], and PST events follow a seasonal pattern, as they are often initiated at the end of spring and reach the termination phase at the beginning of autumn, with an apparent annual consistency [5]. The ability to predict the physical and temporal conditions that will give rise to a PST outbreak with appropriate mathematical-model-based empirical results and regional predictors could protect the local aquaculture and shellfish industry, and, thus, local economies. Thus, in this study we focused on specific patches located within two fishing grounds of the clam *Ameghinomya antiqua* (P.P. King, 1932) to investigate the influence of different environmental variables on PST concentrations over a period of 12 months. Our aim was to identify quantitative field descriptors of PST detoxification at regional and local (fishing grounds) scales.

## 2. Results

### 2.1. Regional Scale

The time series of PSP toxicity based on the concentrations recorded in the fjord and channel system during the *A. catenella* bloom is shown in Figure 1. At the beginning of the monitoring period (December 2008), the toxicity of shellfish samples obtained from Guaitecas Archipelago and Puyuhuapi fjord is below the regulatory limit (80 μg STX eq. 100 g^−1^), while that of samples from Aysén to the Quitralco fjords exceeds the regulatory limit, with values of 80–500 μg STX eq. 100 g^−1^ (Figure 1A). Two months later, toxicity abruptly increases in all areas, reaching a maximum of 21,541 μg STX eq. 100 g^−1^ in samples obtained from Chacabuco channel (Figure 1B). By March 2009, the toxicity of the shellfish samples reaches 1000–20,000 μg STX eq 100 g^−1^, except at some stations located at the Puyuhuapi and Jacaf channels (Figure 1C).

However, in April 2009, toxicity decreases to concentrations between 1000 and 10,000 μg STX eq. 100 g^−1^, except at some stations located at Traiguen Island and Chacabuco channel (Figure 1D). Thereafter, until the end of July, the toxicity gradually declines (80–10,000 μg STX eq. 100 g^−1^) (Figure 1F). By October 2009, the values are between 80 and 1000 μg STX eq. 100 g^−1^ (Figure 1H), and by November 2009 between 80 and 500 μg STX eq. 100 g^−1^ (Figure 1I) and, therefore, still above the regulatory limit.

### 2.2. Harvesting Areas Scale

In summer 2009, an *Alexandrium catenella* bloom was detected in the Aysén region. The highest cells densities (6800 cells L^−1^) were reached on 18 March 2009 in Ovalada Island (Figure 2 and Figure 3A). By April 2009, cells were not present or they were very scarce (<50 cells L^−1^) in both sites, Ovalada Island (Figure 3A) and Low Bay (Figure 2 and Figure 3B). During the summer of 2010, no bloom of *A. catenella* was recorded in the study area.

PSP toxicity values during 2009 were compared at a smaller regional scale from the two Instituto de Fomento Pesquero (IFOP) monitoring stations located at Low Bay and Ovalada Island (Figure 2B,C), as shown in Figure 4. The trend in toxicity is generally similar at the two locations, except that the concentration is six-fold higher at Ovalada Island (maximum of 12,579 μg STX eq. 100 g^−1^) than at Low Bay (maximum of 1347 μg STX eq. 100 g^−1^) between March and May 2009.

The same comparison, but using data gathered from 18 sampling sites (12 at Low Bay and 6 at Ovalada Island; Figure 2B,C) is shown in Figure 5, which provides a more detailed picture of the toxicity distribution within the two zones. The initial toxicity at Ovalada Island during April 2009 is around ten times than that at Low Bay, and in the former it increases from no detection (<30 to μg STX eq. 100 g^−1^) in March to more than 12,000 μg STX eq. 100 g^−1^ in April (Figure 5, upper panel). Moreover, both curves show a two-step profile, characterized by a rapid detoxification phase for up to 6 months, which is appreciably faster at Ovalada Island (steeper drop), followed by a second phase in which the field detoxification rate is much slower as the regulatory limit for human consumption (80 μg STX eq. 100 g^−1^) is approached.

The mean field detoxification rate estimated for Ovalada Island (−0.067 d^−1^) is 20% higher than the one estimated for Low Bay (−0.055 d^−1^). According to the ANOVA results, the observed variability in the field detoxification rates is mainly explained by two geographical variables, zone (31.8%) and patch (13.5%). None of the assessed environmental variables have a relevant explanatory power, as all *p*-values are >0.56 (Table 1). Thus, clam biomass, temperature, salinity, and chlorophyll-*a* together explained just 2.7% of the total variability.

At Ovalada Island, the mean instantaneous toxin field detoxification rate fluctuates between −0.054 and −0.080 d^−1^ (Table 2) and is much higher in the island’s southern patch. However, given the initial PSP toxicity of 12,294 μg STX eq. 100 g^−1^ and the estimated baseline of 142 μg STX eq. 100 g^−1^, no reduction below the safety threshold of 80 μg STX eq. 100 g^−1^ occurs at any time (Table 2, Figure 6). At the northern patch, by contrast, a period of 316 days is required before PSP concentration is considered safe. Within Low Bay, the detoxification rate fluctuates between −0.040 and −0.070 d^−1^, with higher rates in the eastern patch. In the bay, assuming an initial PSP concentration of 675 μg STX eq. 100 g^−1^, the detoxification time required to reach safe concentrations is 92 days (Table 2, Figure 7).

HPLC analyses of *A. antiqua* clams obtained from Ovalada Island on 11 May 2009 (Figure 5, upper panel) demonstrate their high toxicity, with values between 397 and 1621 μg STX eq. 100 g^−1^. The toxin profile (% mol) is dominated by carbamoyl toxins (78.02% of the total toxins) and specifically by GTX2-GTX3 (28.9%), GTX1-GTX4 (20.3%), STX (17.6%), GTX6 (7.09%), and neoSTX (6.97%). Other toxins are detected in smaller amounts and include N-sulfocarbamoyls, such as C3 (9.97%) and C1 (4.26%), and decarbamoyls, such as dcSTX (4.89%). C2 and GTX5 are present at trace concentrations (<1%) (Figure 8).

## 3. Discussion

Biological processes occur over a wide variety of spatial and temporal scales and should be investigated under the same spatio-temporal scales that govern the physical-chemical processes with which they interact [20,21].

In our study of the toxicity of the dinoflagellate species *A. catenella*, we found a significant variability in the spatial and temporal dynamics of PST concentration and field detoxification rates between regions, fishing grounds, and patches. An awareness of these spatio-temporal patterns will allow improved monitoring of biotoxin concentrations associated with HABs [22]. We also determined that, independent of the initial toxicity values (the starting point of the study, marked by the lack of *A. catenella* cells in the water), detoxification followed a two-phase pattern, characterized by an initially fast rate but then, as toxicity approached the regulatory limit for human consumption (80 μg STX eq. 100 g^−1^), a slower rate. This pattern has been reported by other authors [23,24] but was not previously modelled. The mathematical model of detoxification presented in this work can be used to develop predictors of the safety of fishing grounds and, thus, improve their management, such as by minimizing the duration of shellfish extraction bans.

The distribution and density of dinoflagellates are typically not homogeneous, neither at depth nor in surface waters; rather, dinoflagellate occurrence is characterized by a patchiness and heterogeneity that depends on variables such as salinity [25] and small-scale physical and chemical factors (e.g., [26,27,28,29]). The uneven cell distribution likely accounted for the variability in PST concentrations recorded in this study. However, this does not imply that the PST distribution is not predictable, as a study by Díaz, Molinet, Seguel, Díaz, Labra, and Figueroa [12] in the same area found a similar pattern in the density of resting cysts of autotrophic dinoflagellates after a bloom of *A. catenella*. The latter authors postulated the existence of “growing niches” and/or coastal hydrographic conditions favoring cell accumulation and, consequently, cyst deposition [30].

Our study shows large differences in the field detoxification rates at the fishing ground scale, with faster detoxification measured at the Ovalada Island sampling stations than at stations at Low Bay (Figure 2B,C), although the furthest stations at these two sites are separated by only ~6 km. The difference may be associated with the local circulation pattern. The authors of the above cited study conducted drifter surface experiments that revealed significant differences in the local circulation patterns (velocities and trajectories) of the studied fishing zones. The fastest current velocities were found at the more interior, more protected fishing zone of Ovalada Island, where faster field detoxification rates (k = −0.067) were detected as well. This is in contrast to the findings of Kaga et al. [31] in their study of the scallop *Patinopecten yessoensis*, in which lower maximum toxicity concentrations did not lead to faster detoxification. The reasons for this difference are unknown, but environmental variables such as the circulation pattern, temperature, and salinity may affect detoxification rates, mainly during the early rapid phase [32,33]. In any case, these differences between areas must be taken with caution as the time extension of the current study was unsuitable for assessing the effects of the large inter-annual variability in oceanographic and ecological processes known to affect this area.

Elucidation of the field detoxification kinetics of PSP toxins is a challenging task as HABs are ephemeral and can occur and subside in time scales that are smaller than feasible sampling intervals. Thus, observed responses must be seen as the integrated average of many different ecological and physiological processes, occurring at different time scales. While sorting out the effect and relevance of such processes would require a substantial amount of controlled experiments, we trust empirical field detoxification rates are scientifically relevant and potentially useful indices for practical policy- and decision-making purposes. For instance, predicting detoxification periods required to reach the regulatory limit concentration of 80 STX eq. 100 g^−1^ can be used to set up adaptive harvesting bans periods suitable for reorienting fishing, and industrial efforts aimed at reducing economic losses for fishermen and food-processing industries.

Using a one-box model [19] to estimate field detoxification rates, we obtained a good fit between the predicted and observed values and were, thus, able to demonstrate the existence of “extractive windows” allowing the commercial exploitation of clam stocks from the Aysén region. It is important to note that this model followed detoxification when toxic cells were no longer detected in the water, after a *A. catenella* bloom event in the year 2009, and that, therefore, no retoxification was possible.

Safe exploitation of these cultivation areas on basis of cells counts would be expensive and rather ineffective, given the high costs of intensive monitoring in remote waters. Moreover, as toxic cells may appear in the water column at any time of the year, nearly continuous intense monitoring would be required [34]. *A. catenella* blooms are more and more frequent in Chilean waters [16], showing a pattern of biannual outbreaks [5]. Recurrences within lower periods of time have not been reported so far, probably due to different factors, such as the long dormancy period of *A. catenella* resting cysts (3 months [35]), coupled with the low residence times of resting cysts in the sediments [12], both of which reduce the chances of rapid and successful benthic–planktonic shifts.

However, the implementation of a detoxification model in remote areas, such as the Chilean fjords, might reduce these high sampling costs, by optimizing the sampling frequency, considering the lengthy detoxification times (up to 300 days) required during intense PSP events.

## 4. Conclusions

Our study demonstrates the significant variability in the spatial and temporal dynamics of *A.-catenella*-related PSP toxicity within and between fishing grounds. The field detoxification rate is shown to decrease as PST concentrations approach the regulatory limit for human consumption. Mathematical models predicting field detoxification rates reveal the existence of “extractive windows,” thus, demonstrating the utility of such models to improve the management of shellfish grounds by limiting the duration of bans on bivalve extraction.

## 5. Materials and Methods

### 5.1. Study Area

The study area is located in the northern region of the Patagonian fjords and channels (Figure 2) within one of the most extensive estuarine regions in the world [36], with inputs of freshwater from rivers and glaciers [37]. The irregular topography and bathymetry resulting from the small and large islands, channels, sills, rivers, glaciers, etc., give rise to a complex geographic system in which oceanographic research is challenging. The Moraleda Channel is one the main geographic features of the study area and it contributes to the transport of freshwater mainly from Puyuhuapi and Aysén fjords, but also from the San Rafael Lagoon [38,39]. The freshwater supply is largest during spring–summer, owing to the large-volume river discharge due to ice melting. During this time, stratification increases [39] and accounts for the development of the productive season in this region [40,41].

An intense marine current is present at the mouth of the Guafo River, which communicates directly with Corcovado Gulf and Moraleda Channel [42]. In the interior zone of Moraleda Channel, e.g., at the Meninea constriction, a strong current arises from the action of winds and tides in the circulation regime [43]. Tides are mainly semidiurnal and mixed semidiurnal, with amplitudes of ~2 m [39,44]. The wind regime is dominated on average by westerly winds [45,46], but synoptic-scale variability plays an important role in the weather conditions of the region [47]. Low Bay and Ovalada Island are patchy fishing grounds located in the NW portion of this Patagonian fjord system, where the IFOP has recorded annual catches of up to 978 t and 91 t, respectively. In this study, these stocks were sampled for the clam *A. antiqua* (Figure 2). Oceanographically, Low Bay fishing grounds (Figure 2B) are much more open to the ocean than those of the more interiorly located and protected Ovalada Island (Figure 2C).

Two spatial scales were examined in this study. The large- (regional) scale is shown in Figure 2A, including the stations of the IFOP’s monitoring program. The small-scale consists of the two fishing grounds (Figure 2B,C).

### 5.2. Field Sampling

#### 5.2.1. Regional Scale

Each sample consisted of at least 20 individuals (mainly the clam *Aulacomya atra*) of commercial size (>7.0 cm), collected by fishermen by means of hang-gathering. Samples were placed inside a plastic bag, then in a cool box at 10 °C, and transported to the Health Ministry Laboratory. There, PST was extracted from shellfish tissues following the official Association of Official Analytical Chemists (AOAC) method 959.08 [48]. For toxin extraction, 100 g of homogenized raw tissue was mixed with 100 mL of HCl (0.1 N) using a blender and then boiled for 5 min. The sample was cooled at room temperature for 10 min and the pH was corrected to 2–4. The resulting extract was then transferred to a 200 mL volumetric flask and filled up to the 200 mL mark with HCl (0.003 N). Aliquots (1 mL) of the final extract were intraperitoneally injected into three Swiss mice weighing 19–21 g following official AOAC method 959.08, and their death times were recorded. If any mouse died in <5 min, the test was performed again using diluted samples until the time until death was 5–7 min. The toxicity was calculated and expressed as μg STX eq. 100 g^−1^ sample, using Sommer’s Table.

#### 5.2.2. Fishing Grounds Scale

From March 2009 to March 2010, samples of clams for toxin analysis were collected from 12 stations at Low Bay (Figure 2B) and 6 stations at Ovalada Island (Figure 2C). At each station, monthly samples consisting of at least 20 individuals of commercial size (>6 cm) were collected, transported, and stored until their analysis in a mouse bioassay as described above.

For semiquantitative analyses of potentially harmful microalgae and chlorophyll *a* (chl-*a*) measurements, during sampling two water samples were collected from close to the sea bottom (3–5 m) at each sampling station using a 5 L bottle. For microphytoplankton analysis, water samples (5 L) were passed through a PVC cylinder with a 10 µm sieve, to a final volume (to be measured) of around 50 mL and then fixing them with acidic Lugol’s iodine solution [49]. Later, 10 mL of fixed samples were sedimented for cell counts (detection level ~1 cell L^−1^) under an inverted microscope (Olympus CKX41, Olympus, Center Valley, PA, USA) using the method described in Utermöhl [50]. For chl-*a* and phaeopigments, 200 mL of seawater was filtered through Whatman GF/F glass fiber filters in triplicate and immediately frozen (−20 °C) until analyzed by fluorometry. Acetone (90%, *v*/*v*) was used for pigment extraction (Turner Design TD-700), performed according to standard procedures [51]. Temperature, salinity, and depth were recorded at each sampling station using Star-Oddi DST data loggers.

On 11 May 2009, samples for the determination of the clams toxin profiles were collected from Ovalada Island. The sample consisted of 20 individuals of commercial size (>6 cm), collected by fishermen by means of hang-gathering. In the laboratory, the shellfish were washed and PSTs were extracted from their tissues for subsequent HPLC analysis using the post-column derivation method [52]. Chromatography was carried out on a RP-18 (125–4.5 mm) Supelcosil LC-8 (58220U) column using a Perkin–Elmer system with a Series 200 fluorescence detector (excitation 330 nm, emission 390 nm, PerkinElmer, Shelton, CT, USA). PST standards, i.e., saxitoxin (NRC CRM-STX-c), neosaxitoxin (NRC CRM-NEO-c), decarbamoyl saxitoxin (NRC CRM-dcSTX), gonyautoxin 1 and 4 (NRC CRM_GTX1/4-c), and gonyaulatoxin 2 and 3c (NRC CRM GTX 2&3-c), were obtained from the National Research Council (Halifax, NS, Canada). PST concentrations in the samples were quantified by comparisons with the corresponding reference materials (from IMB-NRC, Ottawa, ON, Canada). The total toxicity of each sample was calculated based on equivalency factors (TEFs) [53].

### 5.3. Modeling of Field Detoxification Rates

A first-order exponential decay, previously used to describe field detoxification dynamics [19], provided a rather poor fit of the data. Hence, this basic model was generalized by incorporating two additional parameters: (i) a baseline toxin concentration, which represented non-zero baseline (post bloom) toxin concentrations, and (ii) a power time exponent that allowed for a decreasing detoxification rate over time. Data support for the following four models was assessed using the second-order Akaike information criterion [54].
(1)$$y_{t}=\;y_{max}\cdot e^{-k\cdot t}\;\;$$
(2)$$y_{t}=\;y_{max}\cdot e^{-k\cdot x^{g}}$$
(3)$$y_{t}=\;y_{bl}+\left(\;y_{max}-y_{bl}\right)\;\cdot e^{-k\cdot t}\;\;$$
(4)$$y_{t}=\;y_{bl}+\left(\;y_{max}-y_{bl}\right)\;\cdot e^{-k\cdot x^{g}}\;$$
where *Y_max_* is the maximum PST value determined at each sampling site, *t* is the time in days since *Y_max_* was achieved, *Y_bl_* is the median PST value determined at each sampling site during the last three sampling events, *k* is the site-specific instantaneous decay rate, and *g* is the across-sites power exponent. Both *k* and *g* were estimated for each of the equations (as needed) through least-squared non-linear models, assuming multiplicative errors. Equation (4) was much more informative than the original one-parameter equation (ΔAICc = 190.3) and its probability (AICc-weight) when compared to the other three models was 0.93.

Due to the short duration, low detectability, and poor correlation between blooms and toxicity, we did not use variability in *A. catenella* density to define t_0_. We used, instead, the actual times at which maximum PST values were observed at each site. As these field detoxification rates are modelled as instantaneous (specific) quantities, they mainly depend on previous PST concentrations and, as a result, str quite insensitive to starting values. The opposite is true if linear functions are used instead.

Once site-specific *k* estimates were obtained, a marginal analysis of variance [55] was conducted to determine the explanatory power of several environmental variables for these rates. These variables included fishing ground zone, patch within fishing zone, temperature, salinity, and chl *a* concentration. Neither a major departure from a normal distribution not homoscedasticity assumptions was found after a graphic and semiquantitative analysis of the residuals (Shapiro–Wilks and Breusch–Pagan tests). All analyses were performed using the statistical and programming software R 3.5.1 [56], packages “car”, “ggplot2”, “nlme”, “lme4”, and “cluster”, available through the CRAN repository (www.r-project.org/, accessed on 14 September 2022).

## Figures and Tables

**Figure 1 toxins-14-00786-f001:**
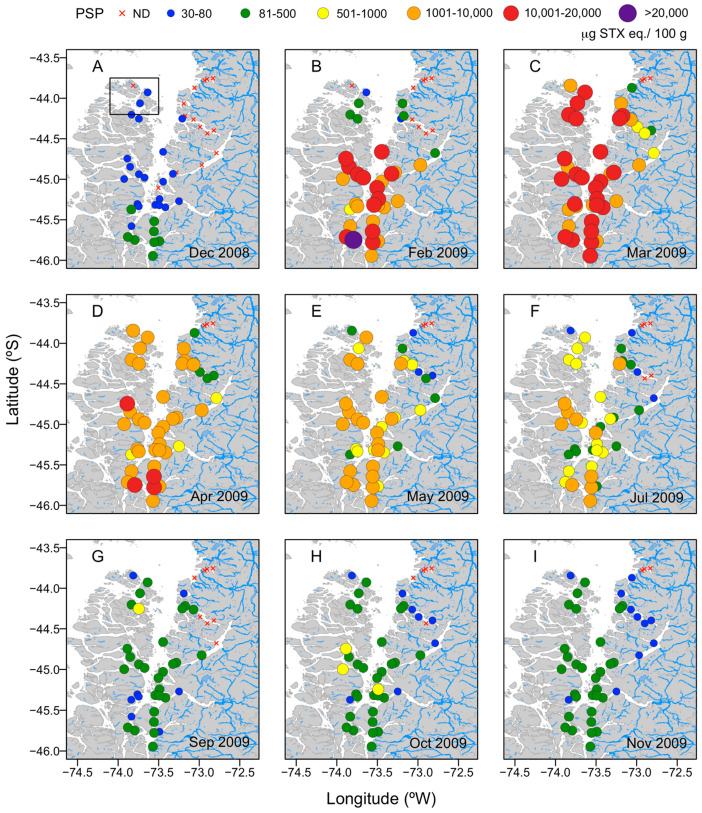
(**A**–**I**) Spatial and temporal distributions of paralytic shellfish poisoning (PSP) toxicity (μg STX eq. 100 g^−1^) in the ribbed mussel *Aulacomya atra* as recorded in a monthly monitoring program carried out in the Aysén region (southern Chile) from December 2008 (**A**) to November 2009 (**I**).

**Figure 2 toxins-14-00786-f002:**
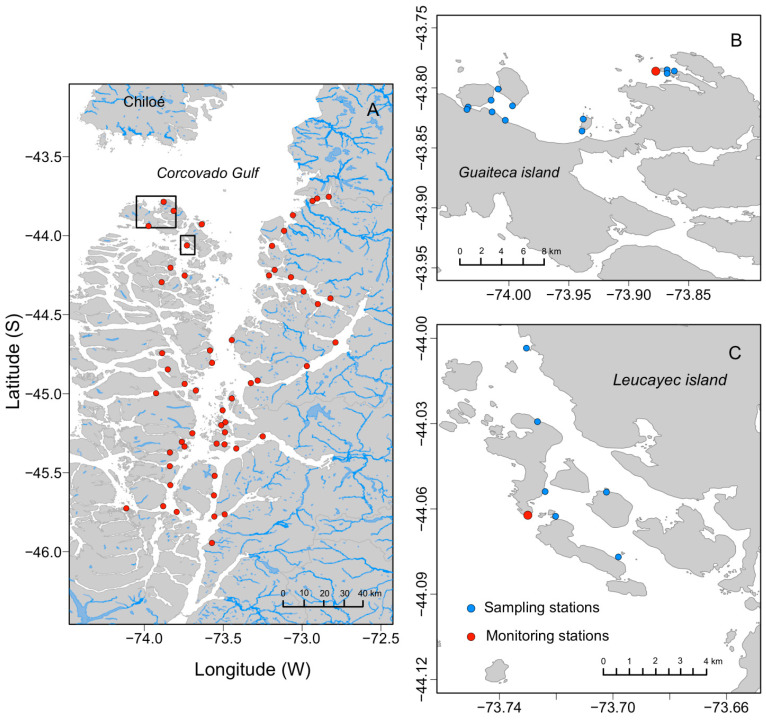
(**A**) Study area in southern Chile (Aysén region), showing the monitoring and sampling stations of the IFOP (red dots). (**B**) The locations of the 12 sampling stations (blue dots) at Low Bay (**B**) and the six sampling stations at Ovalada Island (**C**) are indicated.

**Figure 3 toxins-14-00786-f003:**
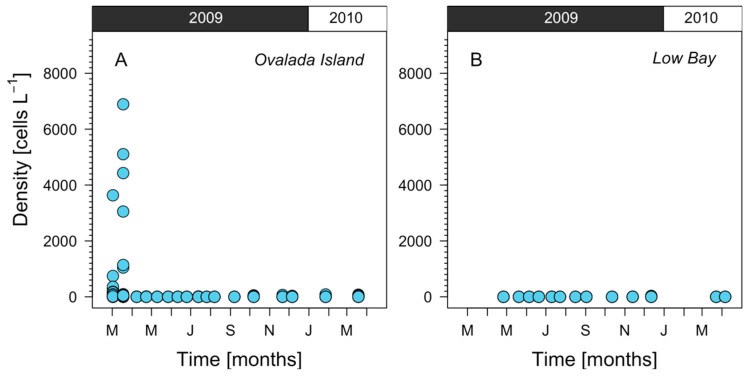
Cell density of *Alexandrium catenella* in the water column detected in (**A**) Ovalada Island and (**B**) Low Bay from March 2009 to March 2010.

**Figure 4 toxins-14-00786-f004:**
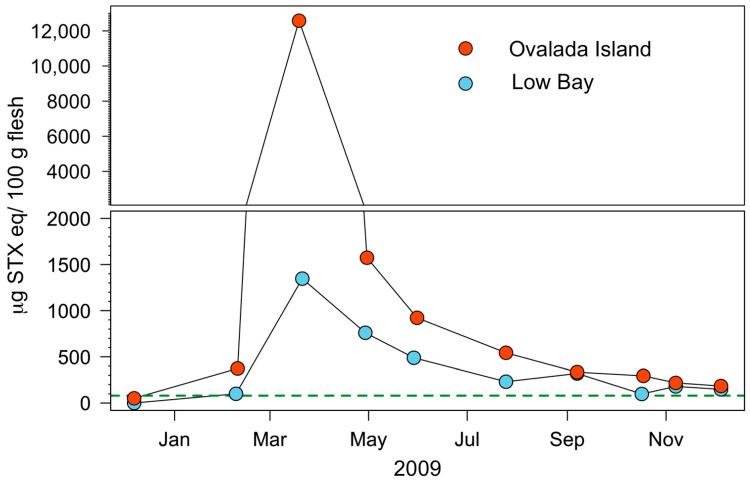
Spatial and temporal distributions of PSP toxicity (μg STX eq. 100 g^−1^) in the clam *Ameghinomya antiqua* recorded at two Instituto de Fomento Pesquero (IFOP) sampling stations (Ovalada Island and Low Bay) during a monthly monitoring program carried out in the Aysén region (southern Chile) from January to December 2009. The green segmented line denotes the threshold of 80 μg STX eq. 100 g^−1^.

**Figure 5 toxins-14-00786-f005:**
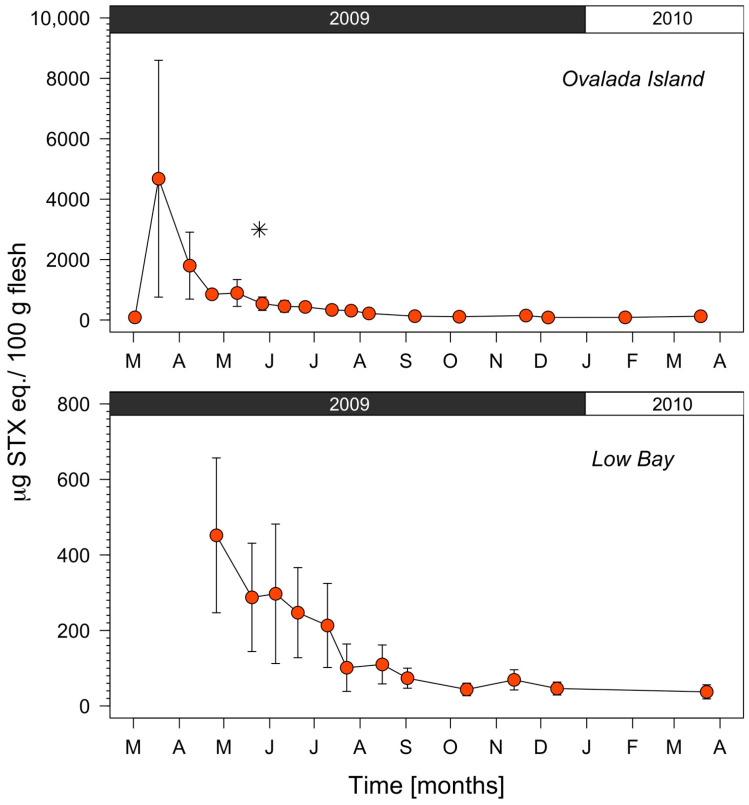
Spatial and temporal distributions of PSP toxicity concentrations (μg STX eq. 100 g^−1^) in the clam *Ameghinomya antiqua* as measured at: (**Upper panel**) Ovalada Island and (**Lower panel**) Low Bay from March 2009 to March 2010. Note the differences in the *y*-axis limits. The asterisk (*) in the upper panel indicates the sample that was analyzed by HPLC, all other samples were analyzed by AOAC official method 959.08. Error bars represent the standard deviation from 6 (upper panel) and 12 (lower panel) clam samples.

**Figure 6 toxins-14-00786-f006:**
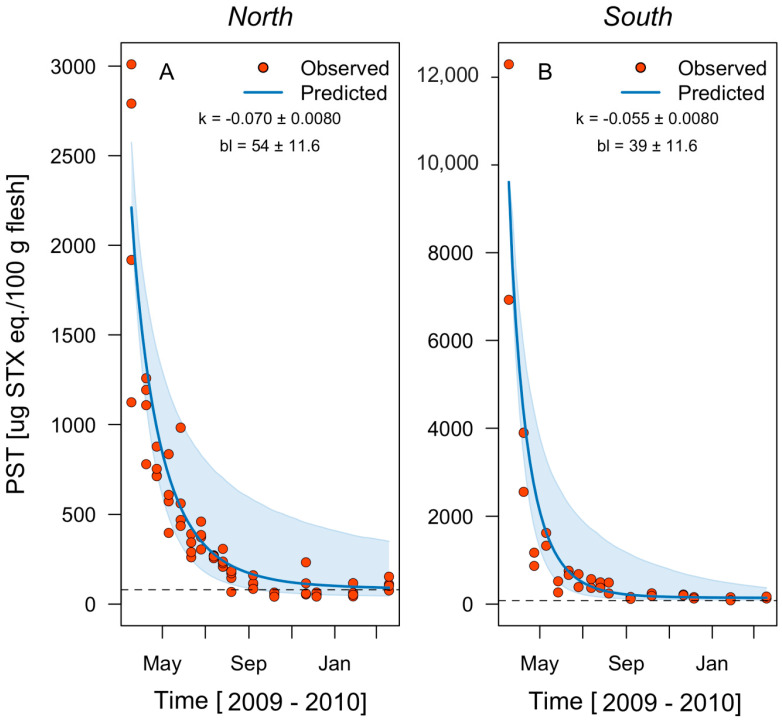
Observed paralytic shellfish toxins (PST) field detoxification rates (red circles) and the rates predicted in a one-box model (blue lines) for the clam *Ameghinomya antiqua* from March 2009 to March 2010 in two patches of the fishing zone at Ovalada Island: (**A**) north and (**B**) south. The instantaneous decay rate (model parameter *k*) is indicated in both cases. Sky-blue shaded areas indicate the confidence interval (95%) for the model estimation, and the dashed lines the regulatory limit for PST consumption by humans (80 μg STX eq. 100 g^−1^).

**Figure 7 toxins-14-00786-f007:**
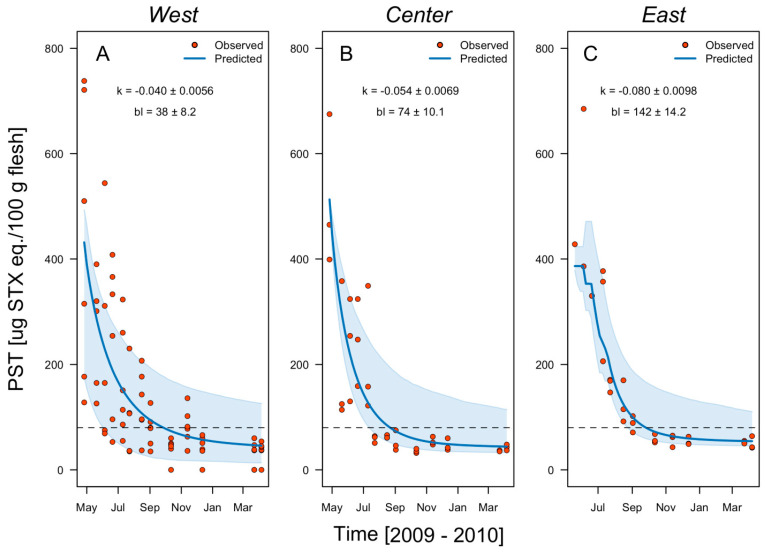
Observed PST field detoxification rates (red circles) and the rates predicted in a one-box model (blue lines) for the clam *Ameghinomya antiqua* from March 2009 to March 2010 in three patches of the fishing zone Low Bay: west, center, and east. The instantaneous decay rate (model parameter k) is indicated in all cases. Sky-blue shaded areas indicate the confidence interval (95%) of the model estimation, and the dashed lines the regulatory limit for PST consumption by humans (80 μg STX eq. 100 g^−1^).

**Figure 8 toxins-14-00786-f008:**
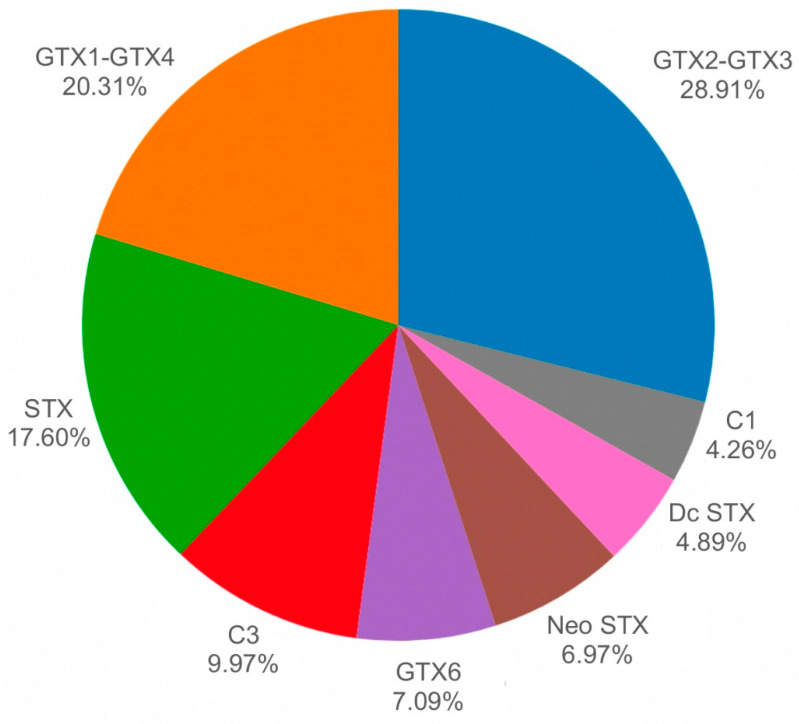
Relative toxin profile (% mol) in whole *A. antiqua* clams collected from Ovalada Island on 11 May 2009.

**Table 1 toxins-14-00786-t001:** Marginal ANOVA of the main effects of fishing zone, patch within zone, salinity, temperature, and chlorophyll-*a* on the field detoxification rates of *Ameghinomya antiqua* as determined at Ovalada Island and Low Bay during the detoxification phase. Significant effects (*p* < 0.05) are showed in bold.

Predictive Variables	Df	SSQ	F Value	*p* (>F)
Null	17	0.005754		
Fishing grounds	1	0.001832	6.977	**0.027**
Zone/patch	3	0.000776	0.986	0.442
Clam biomass	1	0.000096	0.367	0.559
Chlorophyll *a*	1	0.000007	0.029	0.864
Salinity	1	0.000000	0.000	0.998
Temperature	1	0.000051	0.194	0.670
Residuals	10	0.002363		

**Table 2 toxins-14-00786-t002:** Instantaneous decay rate of PST (d^−1^), power exponent, and maximum and minimum PST values estimated/observed for detoxification models of PSTs in the clam *Ameghinomya antiqua* from different patches at Ovalada Island and Low Bay, NW Patagonia. At the southern patch of Ovalada Island, detoxification down to 80 μg STX eq. 100 g^−1^ was not attainable.

Fishing Ground	Patch	K		*g*		PST_max_		PST_min_		Detoxification Time (d)
		Mean	SE	Mean	SE	Mean	SE	Mean	SE	
Ovalada I.	North	−0.054	0.0069	0.82	0.065	2211	331	74	10.1	316
	South	−0.080	0.0098	0.82	0.065	9610	406	142	14.2	NA
	Overall	−0.067	0.0083	0.82	0.065	5910	368	108	12.2	NA
Low Bay	East	−0.070	0.008	0.82	0.065	490	270	54	11.6	92
	Center	−0.055	0.008	0.82	0.065	513	382	39	11.6	103
	West	−0.040	0.0056	0.82	0.065	432	331	38	8.2	136
	Overall	−0.055	0.0072	0.82	0.065	478	328	44	10.5	110

## Data Availability

Not applicable.

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
