# Peer review of "Modelling the Spatial and Temporal Dynamics of Paralytic Shellfish Toxins (PST) at Different Scales: Implications for Research and Management"

_toxins, 2022, doi:10.3390/toxins14110786_

Round 1

Reviewer 1 Report

The authors measured PSTs and phytoplankton in a species of clam from coastal Chile. They performed monthly sampling at a number of sites and calculated detoxification rates from monthly data. The authors calculate detoxification rates from natural populations which has problems, and this reviewer is not convinced that they did due diligence in ensuring their calculations are as accurate as possible.

Overall this paper needs revision – if the authors provide more qa/qc of phytoplankton observations, more years of data if available, and a comprehensive justification that their detoxification rates are calculated using best practices (i.e. throwing out any areas that retoxified), the paper may be considered for acceptance.

Abstract:

What is meant by patchy fishing beds?

This is confusing wording: faster at the less oceanic but more productive and more toxic, not sure what the authors are saying. Maybe restructuring as:

Greater toxicities and faster detoxification were observed at the Ovalada Island site, potentially driven by productivity…?

Introduction

Line 28 eliminate also

Line 28 hard hit is maybe jargon?

Line 29 a overview map would be helpful to orient readers within the country of Chile

Line 32 transferred is probably not the correct word, absorbed or consumed is probably better

Line 49 “The elucidation of regional predictors relevant at a more practical scale would be of enormous value to the respective stakeholders.” – Good point!

Throughout – I suggest concentrations rather than levels

Results -

Figure 1 – include degrees in the S and W

Figure 2 – The x axis should indicate the Year but other text is unnecessary. Suggest showing y axis on log scale, and including 80ug/100g threshold

Materials/Methods

Line 235 – eliminate and thus

Line 235 – glaciers is probably better than ice melt

Line 236 – Not sure why salinity gradient is relevant here

Figure 7 – Please add kilometer scale bar to figures.

Line 273 – Name of the laboratory? Don’t need the word city here.

Line 287 – because of the format of the paper the species has already been identified in the paper

Line 293- “close to the bottom” How deep?

Table 3 is confusing, I’m not really sure what this is meant to convey? Other than data came from two distinct sources?

Results

Figure 6 – Don’t really need the legend since each pie slice is labeled.

I have some problems with the authors methodology that produces the detoxification coefficients.

While it seems that the authors did only begin calculating detoxification rates once they no longer detected phytoplankton in the water, they did not provide any detail about the results of they phytoplankton net tows. How many samples did they see Alexandrium in? I am interpreting that they looked at phytoplankton once a month in conjunction with the shellfish samples, but blooms of Alexandrium are ephemeral and can occur and subside in time scales less than one month. Furthermore, total toxicities in shellfish are not always correlated to Alexandrium cell counts – it’s possible to see high concentrations of PSTs in shellfish with no visible Alexandrium and it is similarly possible to have Alexandrium in the water column with no observed toxicities.

All this to say, detoxification rates are rarely calculated for wild species, and although the author did provide another paper to support this idea Molinet et al. 2010 – they need to provide more support for this methodology. The reason why detox kinetics are hard to observe with monitoring programs is that there is usually a significant amount of time between sampling efforts. In one month toxin concentrations could have changed significantly either up or down in between the two sampling time points, so the calculations of detox rates are somewhat spurious. For instance maybe peak toxicity at Low Bay was a week earlier than sampled and also peaked around 12,000. Detoxification rates would therefore be the same for both regions, so comparisons between sites are not worth much. That the authors then try to explain the variability in detox rates with environmental variables is not recommended with monthly time scales.

See Bricelj and Shumway 1998

“Laboratory studies in which detoxification occurred in a controlled, toxin-free environment, and field studies in which bivalves were transplanted to certifiably toxinfree waters (e.g., Aalvik and Framstad, 1981) obviously provide the most reliable information on detoxification rates.”

Comparisons of total toxicity has and will continue to be done across sites, but generally these reports feature > 1 year of data. Blooms are ephemeral and can move around so only looking at a single year to compare sites is not a good practice. See work from Moore et al. in Puget Sound.

Finally it is not clear to be if the detoxification rates were calculated on a site specific basis. If so it is probably good to show Figure 2 for each site, but again the detoxification rates are probably not precise.

Discussion

I like the modelling approach and agree that it is needed and valuable. It would be valuable if it was performed in a controlled experiment, but understandably that is not always possible. If the authors can provide more support for their field based detoxification calculations it will be a good paper.

Overall the paper needs more QA/QC and method detail - where is the temperature and salinity data? It's included in models but not mentioned here at all. Was it included on daily time scales? Monthly? 

Reviewer 2 Report

The manuscript provides a model to predict PSP detoxification rates for Ameghinomya antiqua in Southern Chile. The manuscript addresses an important question for the management of toxin events in shellfish growing areas which is how to predict toxin concentration in bivalves at different space and time scales. The manuscript has a weakness in relation to the information on the toxic microalgae species which the authors describe in the introduction and methods but do not appear in results. Discussion and conclusions should be improved. The manuscript should be published after major revision.

Line 23: “paralytic shellfish toxins (PSP);”

Replace for: paralytic shellfish toxins (PST);

Add: paralytic shellfish poisoning (PSP);

Line 29: “cially those caused by the dinoflagellate species Alexandrium catenella, responsible for”

Authors should include the taxonomic authority next to the species name the first time the species is cited in the text: Alexandrium catenella (Whedon & Kofoid) Balech

Line 34: “If regulatory limits for PST accumulation in shellfish (80 μg STX eq. 100 g−1) are exceeded”

Replace for: If the regulatory limit for PST accumulation in shellfish (80 μg STX eq. 100 g−1) is exceeded

Line 47: “mation is for the most part lacking and the smallest scales, such as fishing beds, have”

Replace for: mation is for the most part lacking and the smallest scales, such as fishing grounds, have

Lines 57-61:”The ability to predict the physical and temporal conditions that will give rise to a PST 57 outbreak with an appropriate mathematical model based on regional predictors could 58 protect the local aquaculture and shellfish industry, and thus local economies, by restricting the implementation of bans on the extraction of shellfish products to periods and areas when they are truly necessary.”

This paragraph is not well explained, it is stating that at present bans are over implemented without justification but authors are not providing evidences for it. I suggest to erase the last phrase: “, by restricting the implementation of bans on the extraction of shellfish products to periods and areas when they are truly necessary.”

Line 62: patches located within two beds of the clam Ameghinomya antiqua to investigate the in-

Replace beds for fishing grounds or harvesting areas

Include the name of the taxonomic authority next to the species name Ameghinomya antiqua (P. P. King, 1832)

Line 65: stocks at regional and local (fishing bed) scales.

Replace beds for fishing grounds or harvesting areas

Line 90: “Fishing bed scale”

Replace for harvesting areas

Line 91: “PSP toxicity values during 2009 were compared at a smaller regional scale from the”

It is difficult to understand, do you mean between the two monitoring stations?

Line 107: replace meat for flesh in the Y axis

Line 120: In table 1, authors are using “fishing zone”, authors should be consistent with the term used to refer to the fishing ground, harvesting area or fishing zone.

Line 124: In figure 3, authors should replace meat for flesh.

Line 126: The phrase “The asterisk in the upper panel indicates when during sampling the HPLC toxin analyses were performed” is difficult to understand

If only the sample taken on the 11th May was analysed by HPLC and all the other using the biological method, then replace the phrase by: The asterisk in the upper panel indicates the sample that was analysed by HPLC, all the other samples were analysed by AOAC Official Method 959.08.

Line 140: replace meat by flesh

Lines 175-176: In our study of the toxicity of the dinoflagellate species A. catenella, we found a significant variability in the spatial and temporal dynamics of PST levels and detoxification

The authors do not provide evidence about the toxicity of A catenella since they are not providing information on species identification and or its abundance. They should modify the paragraph to describe the focus of the study which is a long PSP event in 2009 associated to a bloom of Alexandrium catenella and provide the reference that demonstrates that this event was associated to this species.

Lines 172-231: Discussion and conclusions should be improved taking into account that authors do not take into account information in phytoplankton species abundance.

Lines 292-297: “For quantitative analyses of potentially harmful microalgae and chlorophyll a (chl-292 a) measurements, during sampling two water samples were collected from close to the 293 sea bottom at each sampling station using a 2.5-L bottle. The microphytoplankton con-294 tent was quantitatively analyzed by filtering the samples through 10-μm Nitex filters 295 (PVC cylindrical collector) and then fixing them with acidic Lugol’s iodine solution for 296 later counting of microphytoplankton.”

Authors describe the method for phytoplankton analysis in the methods section but they do not provide results related to this method. Authors should either include results on phytoplankton analysis or erase the method from the methods section.

Lines 297-298: “The final volume of the concentrate, ~50 mL, was 297 measured to calculate the conversion factor (~50)”

Which conversion factor?

Round 2

Reviewer 1 Report

I have a better understanding of the manuscript now following initial revision and I think the authors did a good job of responding to edits from the initial review. They included text to discuss the limitations of field detoxification rates and provided solid evidence that there was no retoxification during the sampling period. With a better understanding of the manuscript and the goals of the analysis, I think that this paper is worthy of publication with some minor revisions.

I think the authors goal of analyzing field toxin kinetics at different scales (as per the title) would be better served by switching the presentation of the results. If the authors started with broad scale detoxification dynamics (Table 2, Figures 5 and 6) and then proceeded to look at smaller scale dynamics (Figure 3,4) the flow of inquiry would be: what are the broad trends in detoxification we see > do we see those trends replicated on smaller scales > can we determine environmental factors that affect these detox rates on small scales.

In this light, I suggest 2 edits to the small scale analysis which would assist in this comparison between broad scale dynamics and smaller scale dynamics. First, it would seem like samples from IFOP could be grouped in with the rest of the monitoring data from Figures 8b and 8c. It’s not clear to me why the IFOP data are treated separately since they cover the same temporal scale.

Once those data are combined, calculate instantaneous detoxification rates for each site as the authors did for Table 2, but using the data from sites in Figure 4. It may be that the data from the smaller scale models is already included in the broad regional models, but it would be fine to reanalyze these data as a subset of the broader regional models. This would yield two tables of model coefficients which could be compared – do we see more variability in detoxification at small scales (Figure 8b/8c) or larger scales (Figures 5+6)? I think this is a really interesting question which is almost addressed in this paper but should be tackled more directly.

At that point the next question is can we determine what drives the variability in detoxification rates which would lead into the authors modeling of environmental variables, and it appears from their data that environmental data is not a good predictor of detoxification rates which is a fine result and should be included but as it stands it needs a bit more detail on how data are measured (see below).

Small edits

Line 29 – I believe the authors here should not use “hardly” which would imply that Chile has not been affected by HABs.

Line 68 – suggest the “time series” of PSP toxicity….

Line 92 – I would suggest saying “By April 2009 cells were not present…”

Figure 2 – I didn’t notice this before but the single letter month labels are ambiguous, M is March and May and J is July and January. This makes the figure difficult to interpret.

All other figures – don’t really need “flesh” or ‘meat” in the axis labels if you explain in text how shellfish are shucked and homogenized

Why did the authors choose to analyze the IFOP sites independently from the other monitoring stations? They appear to be in roughly the same areas and cover the same temporal scope and species. Why are they treated separately (Figure 2 and Figure 3)?

Figure 5+6 “months on” can be removed from figure axis. Regions defined here should be shown on the map in Figure 8, maybe each region has a different color point since there are callout boxes.

Figure A1 – The axis titles are flipped. Also please add a box in the main panel showing the region of interest from Figure 8.

I think the authors have restructured their look at environmental variables but they are still analyzing temperature salinity and depth. It would be good to see a summary of those data, how often they were collected, etc. Speaking of depth the authors say they collected samples for chlorophyll at 3-5 meters using a 5L bottle, can the authors clarify that they used a Niskin or similar device to ensure they were sampling at depth? How did they get the bottle down to 3-5m?

Some acronyms like AOAC are never defined.

Reviewer 2 Report

The old text and the new text appear together in some parts of the manuscript probably due a failure in the track of changes. Some editing will be needed to correct this. The numbering of the figures needs to be revised since figure 7 is cited before figure 3 for example, it is maybe related to the fact that methods is at the end.

Something that should be changed, since now they are including phytoplankton results, is in line 350 “For quantitative analyses”. The method that they are describing is not quantitative, it is semiquantitative. They are not applying the Utermohl method directly to the water samples, they are filtering the samples and they analysed the content of the mesh. They should change quantitative for semiquantitative, in line 353 as well.

In line 362 please correct the degree sign (−20ºC), it should be ° not º

Apart from that, the authors have addressed all the comments applying the changes that were needed and explaining them in their replies.
